# Measures of Maximal Tactile Pressures during a Sustained Grasp Task Using a TactArray Device Have Satisfactory Reliability and Concurrent Validity in People with Stroke

**DOI:** 10.3390/s23063291

**Published:** 2023-03-20

**Authors:** Urvashy Gopaul, Derek Laver, Leeanne Carey, Thomas Matyas, Paulette van Vliet, Robin Callister

**Affiliations:** 1KITE Research—Toronto Rehabilitation Institute, University Health Network, Toronto, ON M5G 2A2, Canada; 2Human Physiology, School of Biomedical Sciences & Pharmacy, College of Health, Medicine and Wellbeing, University of Newcastle, Callaghan, NSW 2308, Australiarobin.callister@newcastle.edu.au; 3Occupational Therapy, School of Allied Health, Human Services and Sport, La Trobe University, Melbourne Campus, Melbourne, VIC 3086, Australia; 4Neurorehabilitation and Recovery Group, the Florey Institute of Neuroscience and Mental Health, Austin Campus, Heidelberg, VIC 3084, Australia; 5School of Health Sciences, College of Health, Medicine and Wellbeing, University of Newcastle, Newcastle, NSW 2308, Australia

**Keywords:** pressure sensors, tactile, finger forces, sustained grasp, stroke, reliability, concurrent validity

## Abstract

Sensor-based devices can record pressure or force over time during grasping and therefore offer a more comprehensive approach to quantifying grip strength during sustained contractions. The objectives of this study were to investigate the reliability and concurrent validity of measures of maximal tactile pressures and forces during a sustained grasp task using a TactArray device in people with stroke. Participants with stroke (n = 11) performed three trials of sustained maximal grasp over 8 s. Both hands were tested in within- and between-day sessions, with and without vision. Measures of maximal tactile pressures and forces were measured for the complete (8 s) grasp duration and plateau phase (5 s). Tactile measures are reported using the highest value among three trials, the mean of two trials, and the mean of three trials. Reliability was determined using changes in mean, coefficients of variation, and intraclass correlation coefficients (ICCs). Pearson correlation coefficients were used to evaluate concurrent validity. This study found that measures of reliability assessed by changes in means were good, coefficients of variation were good to acceptable, and ICCs were very good for maximal tactile pressures using the average pressure of the mean of three trials over 8 s in the affected hand with and without vision for within-day sessions and without vision for between-day sessions. In the less affected hand, changes in mean were very good, coefficients of variations were acceptable, and ICCs were good to very good for maximal tactile pressures using the average pressure of the mean of three trials over 8 s and 5 s, respectively, in between-day sessions with and without vision. Maximal tactile pressures had moderate correlations with grip strength. The TactArray device demonstrates satisfactory reliability and concurrent validity for measures of maximal tactile pressures in people with stroke.

## 1. Introduction

Loss of tactile somatosensation and motor function impairs the ability of the fingers and the hand to appropriately scale grip force, resulting in deficits in object handling, lifting, and manipulation [1]. Loss of grip strength in the upper limb is one of the most common impairments after stroke, significantly affecting the ability to use the arm and hand in daily functional activities [2,3,4]. Strength deficits in the paretic hand, such as weakness of finger and wrist flexors and extensors [5,6,7], significantly contribute to motor impairments in moderate to severe stroke [3]. Maximal grip strength measurement is associated with upper limb functional deficits and has been shown to have good reliability after stroke (ICC > 0.86) [8,9]. However, this has been questioned in a recent overview of systematic reviews (2004–2014) of upper limb outcomes after stroke, where it was found that grip strength measurements lacked sufficient reliability, validity, and responsiveness to be a robust psychometric and therefore have limited clinical utility [10].

The Jamar hand dynamometer is the most used tool to assess grip strength [11,12,13,14,15]. This dynamometer has good test-retest reliability in people with stroke (ICC 0.80–0.89) [8] and is accepted as the gold standard [16]. However, it lacks responsiveness to strength changes in people with severe loss of grip strength [17] and underestimates the contribution of forces from the finger pads. During a power grip, most of the gripping force on the Jamar hand dynamometer is provided by the extrinsic muscles of the hand, with little contribution from the intrinsic muscles and finger pads [18,19]. However, during grasping and lifting tasks in activities of daily life, objects are commonly handled between the finger pads [20] using the intrinsic muscles. Hence, there is a need for more responsive measures that incorporate a functional grasp that involves finger pads. This will provide for a more functional evaluation of grasp strength deficits and recovery in the upper limb after stroke. 

Tactile pressure is the result of a complex interplay of somatosensory feedback signals and modulated muscle activity in the hand and arm [21,22]. Impaired tactile somatosensation after stroke impairs the ability to discriminate different physical properties of objects, including the extent of friction (slippery versus nonslippery) [23]. The extent of slip between an object and the skin sends tactile cues to correctly adapt the grip force to achieve a successful grasp [1,21]. Impaired discrimination of surface friction contributes to pinch grip deficits after stroke [24]. So far, the control of finger forces during grasping has been poorly addressed in stroke rehabilitation, even though it is a skill necessary for successful object manipulation in everyday life [25,26].

Pressure sensor devices constitute an advanced technology that can be used to evaluate grip strength deficits [27,28] in people with stroke. Advanced sensor-based technologies, such as Interlink FSR^®^ (Tekscan, Inc., Norwood, MA, USA) [29], Peratech QTC™ (Peratech Holdco Limited, Sedgefield, UK) [30], Tactilus^®^ (Sensor Products Inc., Madison, WI, USA) [31], Sensitronics^®^ (Sensitronics LLC, Bow, NH, USA) [32], and the Tekscan grip pressure mapping system (Tekscan, Inc., Norwood, MA, USA) [33], have been used to sense pressure in hand or grip evaluations [34,35,36,37,38,39]. These sensors are based on piezoresistive sensing technology and are stiff and frail, though they have good sensitivity [39,40]. Alternatively, capacitive sensors have been widely used in exploring mechanoreception in human fingers because of their higher sensitivity to tangential finger forces [41]. 

The TactArray pressure distribution system (Pressure Profile System, Los Angeles, CA, USA) [42] is another sensor-based technology that uses highly sensitive capacitive tactile pressure sensors [43]. TactArray sensing technology has been found to be suitable in surgical robots for tissue palpation [44] and in robotic hands to detect slippage in dexterous tasks [45]. Similarly, because of its high sensitivity, TactArray could be suitable for assessing small variations in tactile pressures or forces [46] in stroke research trials. TactArray has excellent reliability in evaluating maximal tactile pressures and forces in healthy people [47]. Even though a TactArray device has been used to evaluate grasp forces in people with moderate to severe stroke [48], no study has investigated the reliability of this device to evaluate grip strength in people with stroke. This study therefore builds on the systematic evaluation of TactArray previously conducted in our laboratory [47] and provides a detailed preliminary investigation to determine its reliability across a range of measures in people with stroke.

The test-retest reliability of an assessment tool can be measured with intraclass correlation coefficients (ICCs), variations in the mean, as well as the magnitude of systematic and random errors [47,49,50]. The number of trials used to evaluate the reliability of maximal isometric grip strength in symptomatic and asymptomatic populations considerably varies depending on whether the maximum value obtained from multiple trials or the mean of two or three trials is used [15,16,51,52,53,54,55]. There is no consensus on the number of trials to be included, particularly during the estimation of the reliability of sustained grasp [56,57,58].

Evaluating grip strength in both hands [15,55] could be useful for characterizing poststroke deficits. A significant reduction in maximal grip force has been observed in the affected hand compared with the less affected hand in people with stroke [9,59]. The forces produced by the fingers of the affected hand are reported to be 36% less than those of the less affected hand in people with stroke [60]. Deficits in grip strength were also observed in the less affected hand poststroke compared with healthy individuals [61]. Given that grip strength deficits on the less affected side could limit the ability to compensate for functional impairment of the more affected upper limb, further studies are required to investigate the bilateral grip strength deficits after stroke. Bilateral assessment of the reliability of handgrip measurements could help improve the interpretation of findings measured from the affected and less affected hands in stroke research trials. 

Evaluations of maximal grip strength have focused on the instantaneous peak force obtained during a short duration isometric contraction up to 3 s to evaluate the muscle weakness of the paretic hands after stroke [55,62,63]. Activity limitation is positively related to muscle weakness and maximal grip strength when maximal strength is sustained for more than 3 s [64] but not for a contraction lasting less than 3 s [64]. Grip dynamometry, to evaluate maximal grip strength over a short duration, does not provide insight into possible variations in grip strength throughout the duration of the grip [62]. Therefore, investigating the grip strength-time profile of a sustained grip would provide useful information on force, during grip formation, sustained grip, and grip release [57,65], which would help understand functional impairments after stroke.

Visual feedback during grip strength assessments may influence the magnitude of force production [66] and the reliability of these assessments. This may particularly be the case for stroke survivors who experience impaired touch sensation [66,67]. In stroke survivors, variations in maximal voluntary contractions in a grip task were reported when evaluated under different visual feedback conditions [68]. When vision was occluded, fluctuations were observed in people with stroke who experience impaired touch sensation, with more irregular or discontinuous force output in those with marked deficits in grip control, whereas a uniform pattern of grip force was maintained during a pinch-lift task in healthy people [67]. It is therefore a priority to determine the reliability of grip strength assessment with and without vision in stroke survivors.

The concurrent validity of a grip measurement device needs to be evaluated against a gold standard [69]. The TactArray device has moderate to large correlations with grip strength in healthy people [47]. However, the validity of the device has not been investigated in people with stroke. Hence, it is a priority to evaluate the concurrent validity of sustained grip strength to determine its clinical relevance among people with stroke.

The primary objectives of this study were to: (1)Assess the test-retest reliability of maximal tactile pressures and forces during a sustained grasp task using a TactArray device and determine which measures of maximal tactile pressures or forces are most reliable in people with stroke;(2)Determine whether the duration over which sustained grasp data are measured influences the reliability of a TactArray device pressure and force measures in people with stroke;(3)Evaluate the concurrent validity of measures of maximal tactile pressures or forces during a sustained grasp using a TactArray device relative to grip strength using the Jamar dynamometer in people with stroke.

The secondary objectives were to:(1)Determine the percentage difference in maximal tactile measures between the affected and less affected hands in people with stroke;(2)Determine whether there are differences in maximal tactile measures under vision and no vision conditions.

## 2. Materials and Methods

### 2.1. Design

Repeated measurements were collected to assess the reliability of measures [70] of maximal tactile pressures and forces using a TactArray device in people with stroke. To determine the reliability of within-day sessions, the participants were evaluated twice on the same day, one hour apart. To evaluate the reliability of between-day sessions, an additional test [49] was performed one week later. All measures were conducted by one assessor. This reliability study was reported according to the guidelines and checklist for Reporting Reliability and Agreement Studies (GRRAS) [70].

### 2.2. Participants

Stroke survivors were recruited through consecutive sampling through hospitals, the Hunter Medical Research Institute research register, and stroke support meetings. Stroke survivors were included if they: (1) had a confirmed diagnosis of stroke; (2) were adults aged 18 years or older; (3) had sufficient voluntary muscle contraction in the paretic upper limb to reach forward; and (4) had sufficient ability to generate the beginning of prehension to grasp a 5 cm wide object, with or without somatosensory deficits. Stroke survivors were excluded if they: (1) had a prior history of central nervous system dysfunction other than stroke; (2) had upper limb deficits resulting from nonstroke pathology; (3) had any peripheral neuropathy in the upper limb; (4) had moderate to severe receptive aphasia (<7 on ‘receptive skills’ of Sheffield Screening Test for Acquired Language Disorders [71]); (5) if they were receiving therapy for the upper limb at the time of the study; and (6) the ability to hold the TactArray device with the affected hand without any assistance. The participants with stroke were screened over the phone to determine initial eligibility. Those who passed the phone screening attended a preclinical visit to determine final eligibility. 

The characteristics of the participants with stroke were based on the following standard clinical measures: the Wolf Motor Function Test (WMFT) [72], Action Research Arm Test (ARAT) [73], Fugl-Meyer Assessment for Upper Extremity (FMA-UE), Box and Block Test (BBT) [74,75], grip strength (Jamar dynamometer, Patterson Medical, Pemulwuy, Australia) [76,77], pulp-to-pulp pinch strength (B & L Engineering, Santa Ana, Canada) [78], Modified Tardieu Scale (MTS) [79], Tactile Discrimination Test (TDT) [80], Stroke Impact Scale (SIS) [81], the Motor Activity Log (MAL) [82,83], and a pain visual analogue scale (PVAS) [84]. Standard objective performance-based neuropsychological tests were also performed: the Montreal Cognitive Assessment (general indicator of cognitive performance) [85,86], the Star Cancellation Test (neglect) [87,88], and the Rey-Osterrieth Complex Figure Test-copy condition (indicator for visuospatial perception) [89]. The clinical and neuropsychological measures were performed at the end of the 2nd assessment session.

All participants provided written informed consent for the study, in line with the Declaration of Helsinki [90]. Ethics approval was granted by the Human Research Ethics Committee of the University of Newcastle, Australia (Reference No: H-2015-0052) and the Hunter New England Human Research Health Committee, Australia (No: 13/12/11/4.02).

### 2.3. Data Collection with TactArray Device

Data were collected during a grasping task function using a TactArray device as described in detail by Gopaul et al. [47]. In brief, the device was custom-built from the commercially available sensors of the TactArray pressure distribution system and consisted of conformable pressure sensor arrays (432 individual pressure sensing units) wrapped around a cylindrical object (5 cm diameter; 12 cm height; mass: 100 g) (TactArray model T4500, Pressure Profile System; Los Angeles, CA, USA) [91] (Figure 1).

#### 2.3.1. Procedure for Assessment of Maximal Tactile Pressures

The TactArray cylinder was placed on a table surface 15 cm from the hand starting position, directly aligned with the hand, with the wrist in a neutral position [91]. Standardized instructions were given to participants to reach, grasp, and lift the TactArray cylinder to a height of 2–5 cm, then to hold and squeeze as hard as they could for an 8 s period [92] using a 5-digit multifinger prehension grasp [93], then place the object back on the table. The object was picked up with the distal pads of the fingers without involving the palm of the hand [47]. 

First, the assessor demonstrated the task, then the participants performed one practice trial with each hand to try the finger positions on the TactArray device using sub-maximal effort. Participants were blinded to their performance results during measurements. No verbal encouragement was provided during the maximal grasp task execution [47].

Maximal tactile pressures/forces were assessed bilaterally. with three consecutive repeated measures for each hand. Ten to fifteen seconds of rest was provided between each measurement trial to minimize fatigue. The measurements were carried out in two conditions: with and without vision [66]. Between the measurements for each condition, one to two minutes rest was provided.

In the first testing session, measures were first performed with the less affected hand. The order of the hand testing was randomized across the other two testing sessions. Testing was first performed with vision for each hand. Testing was performed with alternating hands for each condition [47]. Unsuccessful trials (e.g., unable to achieve a stable grasp; did not sustain grasp for 8 s), trials with clear submaximal effort, and trials with reduced sincerity of effort reported by the participant were discarded [47].

#### 2.3.2. Data Processing

The pressure (kPa) on each active sensor was collected and preprocessed offline to reduce noise using customized MATLAB script (R2015b). Data were only obtained from activated sensors, i.e., those having a nonzero value. Measures of ‘tactile’ pressures, i.e., pressure on the surface of the TactArray device sensors with which the finger is in contact, were obtained using this pressure device. The total tactile pressure values were extracted and then converted into total tactile force values as described previously [47].

#### 2.3.3. Determining Maximal Grasp Measures

For each trial, maximal tactile pressures and forces were calculated over two time-frames: (1) the complete duration of the grasp over 8 s (from finger contact to finger release, and (2) over the middle 5 s of the stationary hold or plateau phase of the grasp [47,94]. The plateau phase was set to start two seconds after the auditory cue to accommodate for finger contact, time to overcome preload forces, and after changes in acceleration during lifting had ceased) [47,92]. 

Maximal tactile pressures and forces are reported using: (1) the highest value among the three repeat trials; (2) the mean of the two repeat trials [47,58,95,96] having the least variation; and (3) the mean of the three repeat trials [47,96]. Table 1 provides a summary of the maximal tactile pressure and force values reported in this study.

### 2.4. Data Analysis

Microsoft Excel (Microsoft Office Professional plus 2013) was used for tabulation of all data, which were then exported into the appropriate analysis programs. Microsoft Excel 2013 and the Statistical Package for the Social Sciences version 24 (SPSS Inc., Chicago, IL, USA) were used to conduct analyses. Descriptive statistics, including means and standard deviations, are reported. 

#### 2.4.1. Test-Retest Reliability Analyses

The test-retest reliability of maximal tactile pressures and forces was separately estimated for each pair of consecutive sessions (within-day and between-day) using a consecutive pairwise analysis [97]. Mean raw scores are reported for each testing session. Measures of reliability were calculated based on the log -transformed data to reduce bias arising from nonuniformity error, which is common with small samples sizes. To evaluate test-retest reliability, the percentage change in mean scores, systematic error, typical error, and intraclass correlation coefficients (ICCs) with 90% confidence intervals were calculated for each estimate of maximal tactile pressures and forces [49,98]. 

#### 2.4.2. Indices of Reliability

Group reproducibility was assessed with the percentage change in mean scores between consecutive sessions [49,99]. Changes in the mean of <5% were interpreted as very good, ≥5% and <10% as good, and ≥10% as unsatisfactory [100]. A statistically significant change in the means of the two consecutive sessions was based on the criteria that the confidence interval did not overlap zero [50,99,101].

Systematic errors were evaluated by examining the slope and intercept of the regression line through a scatterplot of the test versus retest scores and by calculating the average difference (AVdiff) between the test and retest scores [102]. A slope close to 1 with an intercept close to 0 and an AVdiff of 0 indicated small or absent systematic error. When Avdiff was not 0, a Student’s *t*-test was conducted on the log-transformed data to determine the significance of the difference. A nonsignificant difference (*p* > 0.05) indicated only a small systematic error [102]. 

Typical error (s) of the log-transformed data was used to assess the within-subject reproducibility, which is expressed as a coefficient of variation (%CV), according to the formula: $\mathrm{CV}=100\left(e^{s}-1\right)$, where the $\mathrm{typical}\;\mathrm{error}\;\left(s\right)\;\mathrm{in}\;\mathrm{each}\;\mathrm{trial}=s_{diff}/\sqrt{2}\;$, and $s_{diff}$ is the standard deviation of difference scores between trials [49,50]. Values < 5% were considered desirable, <10% were good, ≤15% were acceptable, and >15% were considered unsatisfactory [58,100,103]. We also calculated the smallest detectable change based on the smallest difference between two independent measures that exceeds measurement errors [49,50,97]. 

The ICCs were calculated using a two-way analysis of variance (ANOVA), with two factors: absolute agreement and single rater (ICC 2,1) [49,97]. Reliability from ICCs were interpreted as: very good (>0.9), good (>0.75), and unsatisfactory (<0.5) [100].

#### 2.4.3. Analysis of Concurrent Validity

Concurrent validity based on linear regression [97], standard error of estimates, and Pearson correlations between the log-transformed measures of maximal tactile pressures were evaluated during session 1 (with vision condition) and Jamar-dynamometer-derived grip strength measures in both hands. The magnitude of the correlation was interpreted as: trivial (<0.10), small (0.10–0.29), moderate (0.30–0.49), large (0.50–0.69), very large (0.70–0.89), nearly perfect (0.90–0.99), and perfect (1.00) [104].

#### 2.4.4. Analysis of Variance: Vision and No Vision

Analysis of variance (ANOVA) models were used to analyze data from the repeated measurements to evaluate any differences in values between the vision and no vision conditions in each hand and differences between hands with and without vision using the most reliable maximal tactile measure. Three factors could impact the test performance: visual conditions, the hand side used, and the repetition of testing sessions. Therefore, a 2 (vision) × 2 (side) × 3 (session) ANOVA for repeated measures on all three factors was carried out to quantify the main effects and interactions of these variables. Factor one (with vision and without vision) was called vision; factor two (affected versus less-affected hand) was called side; and factor three (testing session 1 versus testing session 2 versus testing session 3) was called session. A *p*-value < 0.05 was used to interpret statistical significance.

## 3. Results

### 3.1. Characteristics of Participants

This study included 11 participants with stroke. The dominant hand was the hand used for writing or the hand predominantly used when performing a task prestroke [105]. Table 2 summarizes the characteristics of the participants (mean age: 64.1 ± 9.0 years). None of the participants reported pain (pain visual analogue scale). Table 3 and Table 4 summarize the scores on the clinical and neuropsychological measures for the affected and less affected upper limbs in these participants.

### 3.2. Reliability of Values of Maximal Tactile Pressures/Forces in Participants with Stroke

The results for the test-retest reliability of the measures of maximal tactile pressures and forces with vision during the complete grasp duration (8 s) are summarized in Table 5 and Table 6. The results for the test-retest reliability of the measures of tactile pressures and forces during the complete grasp duration (8 s) without vision and during the plateau phase (5 s) with and without vision in both hands are summarized in Appendix A. A summary of the evaluation of the measures of maximal tactile pressures and forces with and without vision during the complete grasp duration (8 s) and during the plateau phase (5 s) in both hands, as evaluated against the reliability criteria descriptors, are summarized in Table 7 and Table 8. 

As seen in Table 7, many of the examined measures had good to very good reliability, as determined by the change in the mean and ICCs, whereas the coefficients of variation were far less often acceptable. The most reliable measure for both within- and between-day assessments with and without vision in both the affected and less-affected hands was the average pressure over 8 s of three trials (Pres(8s)avg3). Other measures with acceptable to very good reliability for between-day assessments were the average pressure (Pres(5s)avg3) and force (Force(5s)avg3) over 5 s of three trials both with and without vision in the affected hand. 

For measures of maximal tactile pressure in the affected hand (Table 7), the results for the changes in the mean were good, coefficients of variation were acceptable, and ICCs were very good for within-day sessions with vision using Pres(8s)avg3. For within-day sessions, the changes in the mean and ICCs were good to very good, but the coefficients of variations were unsatisfactory using all other measures of maximal tactile pressure, whether with or without vision. For between-day sessions, the changes in mean and ICC were very good, and coefficients of variation were acceptable for all the pressure measures with vision but only Pres(8s)avg3 and Pres(5s)avg3 without vision.

For measures of maximal tactile force in the affected hand (Table 7), the reliability results for the changes in the mean were good, coefficients of variation were acceptable, and ICCs very good in within-day sessions using Force(8s)avg3 and Force(5s)avg3 with vision. For between-day sessions, the changes in mean were good, coefficients of variation were acceptable, and ICCs were very good using Force(8s)max, Force(5s)avg2 and Force(5s)avg3 without vision.

As seen in Table 8, many more measures in the less affected hand were reliable when measured without vision than with vision. For the measures of maximal tactile pressure in the less affected hand, the reliability results for the changes in mean were very good, coefficients of variations were acceptable, and ICCs were good to very good using Pres(8s)avg3 for between-day sessions but not within-day session, both with and without vision. The changes in the mean and coefficients of variations were good and those in ICCs very good for within-day sessions without vision using Pres(8s)max, Pres(5s)max, Pres(5s)avg2, and Pres(5s)avg3. For between-day sessions with vision, the changes in mean were very good, coefficients of variations were acceptable, and ICCs were good to very good using Pres(8s)max and Pres(8s)avg2, as well as Pres(8s)avg3. For between-day sessions without vision, all the pressure measures had at least acceptable reliability. 

For the measures of maximal tactile force in the less affected hand (Table 8), the reliability criterion for the changes in mean, coefficients of variations, and ICCs was good to very good for both within-day and between-day sessions without vision using Force(5s)max, Force(5s)avg2, and Force(5s)avg3. For between-day sessions, the changes in mean were good, coefficients of variation were acceptable, and ICCs were very good with vision using Force(8s)avg2 and Force(8s)avg3.

### 3.3. Most Reliable Measures of Maximal Tactile Pressures or Forces

The scatter plots of the test-retest raw scores were visually inspected and indicated that the test performance of one participant with stroke (ID K11S) in the affected hand was an outlier for testing session 3 with vision. After removal of this outlier, the indices of reliability were acceptable to very good using all measures of maximal tactile pressures for between-day sessions with vision. The indices of reliability and reliability criteria are reported without the outlier data in Table 5 and Table 7 and Appendix A. 

The number of indices of reliability that met the reliability criteria was slightly greater for measures of maximal tactile pressures than for measures of maximal tactile forces in both the affected (pressures|forces: 54|51) and less affected (pressures|forces: 52|48) hands. In the affected hand, there was better satisfactory reliability with vision than without vision, whereas in the less affected hand, reliability was greater without vision than with vision for measures of maximal tactile pressures. In the affected hand, reliability was greater for between-day sessions than within-day sessions for measures of maximal tactile pressures. In the less affected hand, reliability was similar for within-day and between-day sessions for measures of maximal tactile pressures. For both hands, reliability was similar during the complete grasp duration (8 s) and during the plateau phase (5 s) for maximal tactile pressures. Using the highest value and the mean of two or three trials, the indices of reliability were satisfactory, though the mean of three trials to estimate maximal tactile pressures had greater reliability than the highest value and the mean of two trials. Therefore, in people with stroke, Pres(8s)avg3 and Pres(5s)avg3 were most reliable for between-day sessions. Subsequently, Pres(8s)avg3 was the measure used for the analysis of systematic error and analysis of validity. These measures were consistent with those in another study on maximal tactile measures using the TactArray device [47] and in other studies on grip strength [9,51].

### 3.4. Systematic Error

The measures of tactile pressure were evaluated for systematic error using the average of three repeat trials during the complete grasp duration (8 s), with or without vision, for within- and between-day sessions. The systematic errors were not statistically significant for either pair of consecutive sessions in both hands with vision (range of systematic error: −9.43, 6.67%; range of *p* values: 0.44, 0.07) or without vision (range of systematic error: −3.73, 8.97%; range of *p* values: 0.19, 0.31). Table 9 summarizes the statistical significance of the Student’s t-test analyses of the difference in means (log-transformed data) between the two consecutive testing occasions. 

### 3.5. Concurrent Validity of Measures of Maximal Tactile Pressures

Pearson correlation analyses (Table 10) showed that measures of maximal tactile pressures in the affected hand had significant moderate correlations with grip strength (r = 0.4, *p* = 0.002). In the less affected hand, moderate correlations were found between maximal tactile pressure and grip strength, though not significant (r = 0.6, *p* = 0.07).

### 3.6. Differences in Maximal Tactile Pressures between Vision Conditions and between Hands

A 2 (vision) × 2 (side) × 3 (session) ANOVA for repeated measures on all three factors indicated no statistically significant effects except for that due to side, i.e., mean maximal tactile pressures were significantly lower when performing with the affected side compared with performing with the less affected side (F(1,10) = 7.94, *p* = 0.02). In addition, significant interactions effects were found with vision, with maximal tactile pressures significantly higher in tests with vision than in those without (F(1,10) = 11.76, *p* = 0.01). There were no interaction effects between side (the hand used) and vision conditions (*p* = 0.24). Higher mean maximal tactile pressures were observed in the less affected side (mean: 348; standard error: 8; CI: 331, 365) compared with the affected side (mean: 332; standard error: 9; CI: 314, 352) in the group with stroke based on log-transformed data. 

## 4. Discussion

### 4.1. Reliability of Measures of Tactile Pressures and Forces

This study evaluated the reliability indices of measures of maximal tactile pressures and forces during sustained grasp task using a TactArray device in people with stroke. The indices of reliability were systematic changes in the means, coefficient of variation, and intraclass correlation coefficient. The maximal tactile pressure obtained from the mean of three repeat trials for the complete (8 s) grasp duration was the most reliable measure for both the affected and less affected hands, for both within-day and between-day sessions, and both with or without vision for people with stroke. The maximal tactile pressure over 5 s and the maximal tactile force over 5 s averaged over three trials also showed good reliability in the affected hand both with and without vision for between-day sessions. There were moderate relationships between the average pressure of the mean of three repeat trials over the complete grasp duration of 8 s (Pres(8s)avg3) values of the TactArray device and the Jamar dynamometer in both the affected and less affected hands. 

The ability of tactile sensors to reliably measure the tactile pressures or forces during grasping in people with stroke has not previously been rigorously investigated. In this study, the coefficients of variation were relatively large and unsatisfactory for measures of maximal tactile forces compared with the higher reliability for measures of maximal tactile pressures. A similar study conducted in our laboratory [47] also found higher reliability of the measures of maximal tactile pressures compared with maximal tactile forces in healthy individuals. It is possible that maximal tactile pressures were more consistent than tactile forces because during multifinger tasks, each finger can compensate for deficits in other fingers, even though the production of individual finger forces may vary with the contact area [106,107]. This study found that the TactArray device is more suitable for tactile pressure measurements than tactile force measurements in people with stroke.

Large differences in the coefficients of variation were observed within the stroke group across the testing sessions and vision conditions, indicating inconsistent responses from the participants. In people with stroke with impaired hand function, coefficients of variation were smaller (without outlier) in between-day sessions than in within-day sessions using Pres(8s)avg3 in the affected and less affected hand, both with and without vision. The differences in the coefficients of variations between the consecutive testing sessions were smaller without vision than with vision in both the affected and less affected hand. In the less affected hand, the differences in the coefficients of variation were smaller in between-day sessions than in within-day sessions using Pres(8s)avg3 with vision. These differences in coefficients of variations could have been due to the lack of ability to perform the task in a consistent manner due to deficits in grasp performance in the affected hand compared with the less affected hand. These observations suggest that in people with stroke, evaluating both hands without vision could provide more comparable measures for within-day and between-day sessions.

This study found that some measures of maximal tactile pressures had very good ICCs but unsatisfactory coefficients of variations, such as in the affected hand using Pres(8s)max and Pres(8s)avg2 (with and without vision). Similar findings were observed in a previous study conducted in our laboratory investigating the reliability of maximal tactile measures in healthy individuals [47]. This could be due to sample heterogeneity, which could have yielded high ICC values even if the within-subject variation was large [103]. This implies that two sets of data could be highly correlated but not providing consistent values, and this error would not be detected by the ICC. Therefore, the typical error and the coefficient of variation could be better measures of reliability because they are independent of where the individuals rank in a sample, unlike the ICC [103]. This study therefore indicates that it could be beneficial to provide additional measures of reliability using absolute estimates of reliability such as the percentage change in mean and the typical error to prevent erroneous estimation of reliability [49,50,103], as reinforced by other reliability studies [108,109,110,111,112]. Accordingly, a participant who demonstrates a percentage change in the magnitude of maximal tactile pressure that is greater than the percent coefficient of variation is viewed as demonstrating change. For example, in people with stroke, if the intention is to use measures of maximal tactile pressures for the affected hand, using the average pressure of the mean of three repeat trials over complete grasp duration of 8 s, before and after an intervention, the postintervention change needs to be greater than 9.5% when assessed with vision (Table 5) and 14.7% without vision (Appendix A) to be considered as a true change. Hence, the coefficients of variations could facilitate identification of true responders.

In this study, there were no significant changes in the mean for any measures of maximal tactile pressures in people with stroke in either hand for within-day or between-day sessions, suggesting no significant learning or fatigue effects. This was surprising as one might have expected some fatigue effects in stroke survivors, particularly for the within-day assessments. These findings were aligned with those of our study investigating the reliability of maximal tactile pressures amongst healthy individuals [47]. It is noteworthy that changes in mean incorporate random and nonrandom changes that can cause variations in the mean value between two consecutive testing sessions. The random change in the mean accounts for random errors of measurement, such as effects of fatigue, while the nonrandom changes in the mean account for systematic changes, such as a learning effect [49,113]. The findings of this study imply that this testing protocol using the TactArray device was suitable for people with stroke with little or no confounding effect of fatigue. 

The smallest detectable changes observed in this study were smaller than the coefficients of variation, similar to the findings in our previous study amongst healthy adults [47]. This study emphasizes that when the smallest detectable changes are smaller than the coefficients of variation, any change larger than the coefficient of variation should be interpreted as a meaningful change in maximal tactile pressures. Future studies of the TactArray device currently under investigation could evaluate the magnitudes of the smallest detectable changes and the coefficients of variation in a larger sample, and more testing sessions may be required to reduce the extent of measurement error amongst people with stroke.

Within- and between-day reliability can determine the applicability of performance measures in observational or interventional studies. This study found a better acceptable standard of reliability of measures of maximal tactile pressures using the TactArray device with between-day sessions than with within-day sessions. Therefore, this study suggests that measurements of maximal tactile pressures using the TactArray device could be valuable in evaluating the pre- and post-performance measures to determine the effects of an intervention and, to a lesser extent, evaluating performance measures at one timepoint only as in observational studies amongst people with stroke. However, in our study with healthy individuals, preceding this one, we found satisfactory reliability of measures of maximal tactile pressures using the TactArray device for both within-day and between-day sessions, suggesting the appropriateness of those measures in observational studies as well as to determine intervention effects in healthy populations [47]. The limited reliability of the measures of maximal tactile pressures in within-day sessions in people with stroke could be due to impairments in tactile somatosensation and selectivity of motor control in terms of the lack of independent force production and the lack of synchronization of multifinger force production during a finger force production task [114]. In turn, this could lead to a lack of consistency in grip force production [114], which could possibly be even more apparent when measures are taken too close to each other [115]. 

The reliability of the measures of maximal tactile pressures using a TactArray device are affected by experimental parameters such as the estimate of maximal grasp, side tested, and the visual conditions, as demonstrated by our previous study amongst healthy individuals [47]. Similarly, in the current study, the experimental parameters affected the reliability of the measures amongst people with stroke, as reported in Table 5, Table 6, Table 7, Table 8, Table 9 and Table 10 and Appendix A. For instance, while the test-retest reliability measured by ICCs was similar or smaller for within-day sessions than for between-day sessions in both hands with and without vision using Pres(8s)avg3, the differences in the magnitude of ICCs were not particularly evident between hands or between visual conditions. The only exception was in the less affected hand with vision, where the ICCs for within-day sessions were smaller than in the other measures. The ICCs in within-day sessions were lower than in between-day sessions, which could have been due to the participants’ performance being less consistent in within-day sessions than in between-day sessions. This, in turn, could relate to increased measurement errors over the repeated measures in within-day sessions. Alternately, it is possible that the lack of difference in test-retest reliability (measured by ICCs) between hands in people with stroke resulted from bilateral grip strength deficits [61], such that performance measures were inconsistent in both hands, leading to increased measurement errors over the repeated measurements in both hands, but those errors could have nullified each other. 

Several studies have emphasized the need to assess other aspects of grip strength in addition to instantaneous maximum voluntary contractions [56,116]. This study addresses this need for new outcome measures evaluating sustained grasp performance to quantify grasp deficits after stroke. The maximal tactile pressures obtained using the TactArray device during complete grasp duration and during the plateau phase were reliable in the affected hand of people with stroke. Similarly, the reliability of the maximal tactile measures using the TactArray device during the plateau phase of a sustained grasp was previously demonstrated in healthy individuals in our laboratory [47]. The findings from this study highlight the importance of evaluating sustained grasp performance over complete grasp duration rather than instantaneous grasp measures. 

The findings of this study indicate that the average pressure of the three repeat trials during the complete grasp duration (8 s) and during the plateau phase (5 s) for maximal tactile pressures in between-day sessions were most reliable in people with stroke. The use of the average of three repeat trials was consistent with that in our previous study on maximal tactile measures using the TactArray device in healthy individuals [47] and that in other studies on grip strength using handheld dynamometers in symptomatic and asymptomatic individuals [15,16,51,52,53,54,55]. Additionally, the number of trials can influence the estimation of the magnitude of maximal tactile pressures. For example, the measures of maximal tactile pressures using the highest value of average pressure amongst the three repeat trials over the complete grasp duration of 8 s (Pres(8s)max) were higher (5.9%) than when using the average pressure of the mean of three repeat trials over the complete grasp duration of 8 s (Pres(8s)avg3) in the less affected hand with vision. It is therefore recommended that studies report the method used to estimate tactile pressures using the TactArray device.

### 4.2. Concurrent Validity of Measures of Maximal Tactile Pressures

This study found that the maximal tactile pressures measured with the average pressure of three repeat trials during the complete grasp duration with vision had a moderate relationship with grip strength as assessed with a Jamar dynamometer in the affected hand. Additionally, a moderate correlation between these two measures was found in the less affected hand. This study suggests acceptable concurrent validity in the affected hand. These findings suggest that measures of maximal tactile pressures using the TactArray device could be useful in measuring grasp strength in the affected hand.

### 4.3. Impact of Testing with and without Vision in Both Hands

Visual feedback can influence force production during grasping. This study found significantly larger magnitudes of maximal tactile pressures with vision compared with without vision conditions in people with stroke. These findings support those of previous studies that reported significantly larger force production using sensor-based devices with vision in healthy individuals [46,115]. This could be because the absence of visual feedback can lead to the absence of visuomotor corrections that amplify force production errors during isometric contractions due to altered activations of the small muscles of the hand [115]. Additionally, the impact of the absence of visual feedback on force variability is even more apparent in older adults than younger individuals due to the interactive effects between vision and aging [117]. This study highlights the importance of controlling the visual conditions during the evaluation of task performance due to the contribution of vision in compensating for the lack of somatosensation amongst people with stroke, especially amongst older stroke survivors. Further studies are required to investigate the impact of visual feedback on grasp force production in people with stroke.

### 4.4. Implications for Research and Clinical Practice

This study was the first time that the reliability of the measures of the TactArray device was evaluated in people with stroke. The TactArray device provides reliable within-day and between-day measures of the maximal tactile pressures in people with stroke. Therefore, it could be used to monitor incremental changes in grasp performance in stroke survivors during a rehabilitation session as well as to evaluate participants’ or patients’ responses to the effect of an intervention over a period of time. Furthermore, the development of the measures of maximal tactile pressures using the TactArray pressure distribution system and the evaluation of their psychometric properties are in line with the recommendations of the Stroke Recovery and Rehabilitation Roundtable, which included measures of finger individuation, and pinch and grip strength for the evaluation of behavioral restitution [118].

The pressure-time data of the TactArray device can be further explored to provide additional grasp strength measurements in addition to instantaneous peak pressure, which could facilitate the identification and quantification of deficits during the grasp formation, sustained grasp, and grasp-release phases of a sustained grasp in the paretic hand after stroke. Hence, the analysis pressure-time data during a sustained grasp could be useful in characterizing motor or functional limitations after stroke [64,65]. The maximal tactile pressures assessed using the TactArray device could provide a novel means of objectively quantifying grasp strength.

The TactArray device has the advantage of capturing forces of individuated fingers in real time during the functional grasp of an object. Importantly, the sensor is placed on the object rather than restricting tactile sensing of the fingertips. This is likely important when we consider adaptive grasp forces during the exploration and manipulation of an object, especially when the texture and friction characteristics of an object may vary [119]. Better knowledge of the distribution of contact forces in the human hand in grasping tasks is also necessary, which can be achieved by the TactArray device.

The planning and execution of hand function rely on the complex interactions between somatosensation, motor control, and appropriately modulated grip forces during object manipulation [120]. Thus, the analysis of the measures of tactile pressures or forces from the TactArray device provides critical insight onto our ability to interact with objects through touch to gain better understanding of the functional interactions between somatosensory pathways in the brain and motor control. Additionally, even if it is known that the nonparetic upper extremity is also impaired, existing clinical measurement tools are often unable to identify those deficits because of their lack of sensitivity. This study showed that the TactArray device could also be useful in identifying subtle changes in the less affected hand after stroke. Therefore, TactArray sensors could help in the design of personalized and targeted rehabilitation interventions for grasp deficits targeting both hands. 

The grasp strength data provided by TactArray sensors could be useful in informing decisions in clinical practice and in cross-sectional studies investigating the effectiveness of stroke rehabilitation interventions. We demonstrated the value of using the TactArray device to evaluate changes in grasp strength during a Phase 2 clinical trial involving stroke survivors. In the same study, we used the TactArray device as a tool for training graded control of grasp forces as well as control of finger force contributions. The TactArray data were able to help identify deficits in the different phases of grasp as well as subtle gains that were not captured by less sensitive clinical measures. Thus, this study emphasizes the importance of the TactArray device as a sensitive outcome measure for hand and finger function.

### 4.5. Limitations 

The coefficient of variation, derived from the typical errors, reflects the variability in the scores of an individual participant from one testing session to another. In this study, none of the tactile measures met the recommended target of the coefficient of variation (<5% [49,50]), as the coefficients of variation ranged from 9.52 to 14.72% for both hands for between-day sessions using the average pressure of the mean of three repeat trials over the complete grasp duration of 8 s (Pres(8s)avg3). It could be argued that this was due to low reproducibility of force production tasks resulting from impairments in motor performance after stroke [114]. Alternatively, it is possible for the coefficients of variation to be >5% because the trial-to-trial variability was not accounted for when estimating the maximal tactile measures as reported in our previous study in healthy individuals [47]. This study highlights the need to ensure the capacity of an individual with stroke to perform constant maximal tactile pressures/forces. Given that fluctuations in functional performance over short intervals have been reported after stroke [26,121,122], setting the coefficient of variation at <5% could be too stringent and not realistic for people with stroke. Therefore, to improve the reproducibility of maximal tactile measures in repeated trials within a testing session, five repeated trials are recommended, or additional trials are performed until two trials within 10% of each other are obtained. Sufficient rest between trials is required to reduce the effects of fatigue.

It is also likely that the coefficients of variation were larger than 5% because the testing procedure was partially limited by the order effect due to the randomization of the order of hand testing, even though trials with vision were always conducted first. Similar observations were also reported in our study in healthy individuals [47]. While the randomization of hand order testing may reduce the carry-over effects due to learning or fatigue [62], it may compromise the reliability of performance, particularly in a population with stroke where task performance with the less affected hand first could serve as an appropriate form of learning to facilitate task performance with the affected hand. Hence, it is suggested that future studies evaluating the psychometric properties of tactile measures use the same order of hand testing and visual feedback across all testing sessions to reduce inconsistencies in the testing procedure.

This study reported some preliminary findings on the reliability of measures of tactile pressures across a range of measurements and limited external validity, as it does not meet the recommended sample size of 50 participants [50]. Additionally, the evaluations during the three testing sessions did not optimally reduce measurement errors in the tactile pressures. Additionally, this study was limited to a subset of people with mild to moderate impairments in grasp function in the chronic phase after stroke. The sensor data collected in this study required detailed preprocessing to obtain tactile measures during sustained grasping, which could limit their use in clinical practice.

### 4.6. Recommendations for Future Trials

It is recommended for this study to be replicated in a large sample size with sufficient statistical power with at least four testing sessions. It is also necessary to confirm the reliability of tactile measures in people with more severe deficits after stroke. Further studies with sufficient statistical power are required to explore the differences in maximal tactile pressures between healthy people and stroke survivors. The TactArray sensor data of people with stroke could be compared with the normative data of age-matched healthy controls to help characterize deficits in grasp performance in stroke survivors. It would be valuable to evaluate other psychometric properties such as the floor and ceiling effects as well as the responsiveness of maximal tactile pressures in people with stroke. To further increase the clinical utility of maximal tactile pressures, future studies could evaluate the minimal clinically important differences for maximal tactile pressures in people with stroke. Additionally, the construct validity could be evaluated between the measures of maximal tactile pressures and other common gold standard upper limb assessments such as the Wolf Motor Function Test and the Box and Block test. It would also be useful for future trials to evaluate the reliability of low-level tactile pressures/forces, as the amount of pressure/force and the type of muscle contraction could influence the reproducibility of repeated trials. This study could be extended to other neurological conditions that often involve the gradual weakening of grasp strength, such as multiple sclerosis.

## 5. Conclusions

The TactArray device demonstrates satisfactory reliability for measures of maximal tactile pressures during sustained grasp for within-day and between-day testing sessions using an average of three trials with or without vision in people with stroke. Concurrent validity is satisfactory relative to grip strength assessed using the Jamar dynamometer in the affected hand. Maximal tactile pressures can provide a novel means of objectively quantifying sustained grasp strength, which can be further explored in larger trials. 

## Figures and Tables

**Figure 1 sensors-23-03291-f001:**
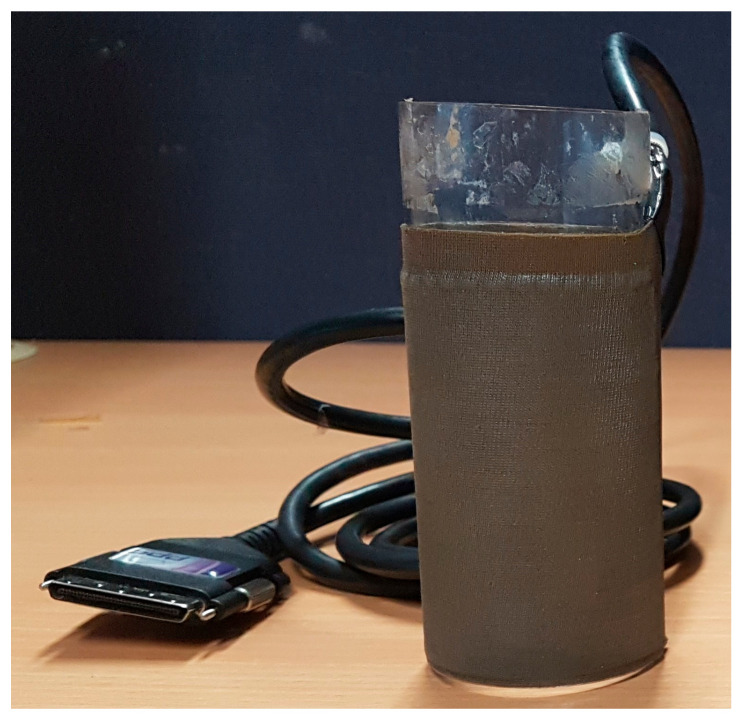
TactArray device.

**Table 1 sensors-23-03291-t001:** Summary of maximal grasp measures and abbreviations.

Variables	Grasp Phase	Number of Trials	Definitions of Maximal Grasp Measures	Abbreviations
Pressure	8 s	Highest value	Highest value of average pressure amongst the three repeat trials over complete grasp duration of 8 s	Pres(8s)max
		Mean of two trials	Average pressure of the mean of two repeat trials having least variation over complete grasp duration of 8 s	Pres(8s)avg2
		Mean of three trials	Average pressure of the mean of three repeat trials over complete grasp duration of 8 s	Pres(8s)avg3
	5 s	Highest value	Highest value of average pressure amongst the three repeat trials over plateau phase of 5 s	Pres(5s)max
		Mean of two trials	Average pressure of the mean of two repeat trials having least variation over plateau phase of 5 s	Pres(5s)avg2
		Mean of three trials	Average pressure of the mean of three repeat trials over plateau phase of 5 s	Pres(5s)avg3
Force	8 s	Highest value	Highest value of average force amongst the three repeat trials over complete grasp duration of 8 s	Force(8s)max
		Mean of two trials	Average force of the mean of two repeat trials having least variation over complete grasp duration of 8 s	Force(8s)avg2
		Mean of three trials	Average force of the mean of three repeat trials over complete grasp duration of 8 s	Force(8s)avg3
	5 s	Highest value	Highest value of average force amongst the three repeat trials over plateau phase of 5 s	Force(5s)max
		Mean of two trials	Average force of the mean of two repeat trials having least variation over plateau phase of 5 s	Force(5s)avg2
		Mean of three trials	Average force of the mean of three repeat trials over plateau phase of 5 s	Force(5s)avg3

s: seconds.

**Table 2 sensors-23-03291-t002:** Characteristics of participants.

ID	Sex (M/F)	Age (y)	Hand Dominance (R/L)	Paretic Side (R/L)	Time Since Stroke (Mo)	Type of Stroke (Isch/Haem)	MOCA	SCT	RFCT	MTS Elbow V1:V2:V3	MTS Wrist V1:V2:V3	MTS Fingers V1:V2:V3
A1S	M	66.2	R	L	79	Haem	23	53	34.5	0:0:0	0:0:0	0:0:0
B2S	M	66.6	R	L	43	Isch	25	54	31	0:0:0	0:0:0	0:0:0
C3S	M	59.3	R	L	224	Haem	23	53	18.5	0:0:0	0:1:1	0:0:1
D4S	F	68.4	R	L	40	Isch	24	53	33	0:0:0	0:0:1	0:0:0
E5S	F	77.0	R	R	24	Haem	23	53	33	0:0:0	0:0:0	0:0:0
F6S	M	63.7	R	R	137	Isch	30	54	34	0:1:2	0:0:0	0:0:0
G7S	F	46.3	R	R	47	Isch	28	54	36	0:0:0	0:0:0	0:0:0
H8S	M	64.9	L	R	185	Isch	24	54	35	0:0:0	0:0:0	0:0:0
I9S	M	70.4	R	R	76	Isch	25	54	36	0:0:0	0:0:0	0:0:0
J10S	M	50.6	R	L	124	Isch	23	54	35	1:1:2	0:0:0	0:0:0
K11S	F	71.4	R	R	79	Isch	16	54	32	0:0:0	0:0:0	0:0:0

M: male; F; female; y: years; R: right; L: left; Mo: months; Isch: ischemia; Haem: hemorrhagic; MOCA: Montreal Cognitive Assessment (maximum score:30); SCT: star cancellation test (maximum score:54); RFCT: Rey Figure Copy Test (maximum score: 36); MTS: Modified Tardieu Scale.

**Table 3 sensors-23-03291-t003:** Clinical measures of the affected upper limb of participants.

ID	WMFT Score	WMFT Time/s	ARAT	BBT *	Grip Strength ^#^	Pinch Strength *	FMA-UE Total Motor Score	FMA-UE Total Sensory Score	TDT Deficit Range Score	SIS Strength	SIS Memory	SIS Emotion	SIS Communication	SISADL	SIS Mobility	SIS Hand Function	SIS Participation	SIS Stroke Recovery	MAL How Much	MAL How Well
A1S	63	50.5	41	34.0	30.0	10.0	66	12	−5.8	20	34.3	55.6	22.9	56	51.1	24	30	50	1.1	1.3
B2S	78	38.9	56	37.3	32.0	9.3	66	11	−11.4	60	45.7	42.2	68.6	62	77.8	36	45	60	1.6	1.8
C3S	41	168.0	30	15.3	18.7	7.3	44	9	−198	40	57.1	53.3	77.1	32	31.1	4	57.5	40	0.7	0.4
D4S	72	42.1	50	28.3	9.3	4.7	62	12	−83.8	40	60.0	60.0	71.4	74	57.8	56	52.5	80	1.8	1.7
E5S	76	43.9	51	36.7	17.3	8.0	66	12	−57.8	75	62.9	68.9	65.7	40	75.6	40	67.5	80	3.5	3.2
F6S	38	92.2	20	16.7	16.7	6.7	42	12	−10.4	55	77.1	55.6	68.6	64	57.8	40	75	70	2.2	2.2
G7S	80	38.7	56	45.3	29.7	9.3	66	12	74.8	50	57.1	66.7	54.3	48	57.8	52	60	75	4.7	4.1
H8S	80	33.5	56	52.7	27.7	9.0	65	12	−12.3	55	71.4	64.4	71.4	76	77.8	76	57.5	80	4.5	4.7
I9S	80	39.3	57	37.3	17.7	7.3	64	12	71.8	20	74.3	60.0	62.9	68	57.8	28	65	60	5.0	3.2
J10S	57	55.1	56	45.0	26.0	5.7	61	12	−4.9	15	65.7	53.3	60.0	6	73.3	56	62.5	60	1.8	2.0
K11S	79	40.2	57	46.0	18.0	6.0	66	12	−81.9	35	2.9	28.9	20.0	72	55.6	56	42.5	80	3.9	4.3

WMFT: Wolf Motor Function Test (maximum score: 75 point); ARAT: Action Research Arm Test (maximum score: 57 points); BBT: Box and Block Test; FMA-UE: Fugl-Meyer Assessment for Upper Extremity (maximum score 66 points); TDT: Tactile Discrimination Test; SIS: Stroke Impact Scale (maximum domain score: 100); MAL-AS: Motor Activity Log-Amount Scale (maximum mean score: 5); MAL-HW: Motor Activity Log-How Well (maximum mean score: 5); ^#^ Mean of three trials using Jamar dynamometer; * Mean of three trials using B & L Engineering pinch grip dynamometer.

**Table 4 sensors-23-03291-t004:** Clinical measures of the less affected upper limb of participants.

ID	BBT *	Grip Strength ^#^	Pinch Strength *
A1S	51.8	39.7	8.4
B2S	50.1	38.3	8.0
C3S	48.9	26.3	7.9
D4S	49.8	14.7	7.8
E5S	50.0	26.0	7.9
F6S	47.7	20.0	7.7
G7S	53.2	33.3	7.6
H8S	53.8	30.0	7.3
I9S	53.0	16.7	6.3
J10S	50.5	36.7	5.5
K11S	46.0	15.3	5.0

BBT: Box and Block test; ^#^ Mean of three trials using Jamar dynamometer; * Mean of three trials using B & L Engineering pinch grip dynamometer.

**Table 5 sensors-23-03291-t005:** Measures of reliability in the affected hand of participants with vision during complete grasp duration (8 s).

		Session 1	Session 2	Session 3	Mean 3 Sessions		Change in Mean (%)	90% CI for Change in Mean	CV (%)	90% CI for Change in CV	Smallest Effect (%) *	90% CI for Change in Smallest Effect	ICC	90% CI for Change in ICC
*Pressure (kPa)*														
Pres(8s)max	Mean	31.29	34.62	33.25	33.05	Session 2-1	9.30	−4.51, 25.12	19.11	13.80, 32.12	10.01	3.96, 13.79	0.91	0.74, 0.97
	SD	11.89	12.24	11.14	11.76	Session 3-2	0.59	−17.78, 23.06	29.81	21.27, 51.54	7.68	−2.31, 11.32	0.71	0.34, 0.89
						Session 3-2 ^#^	−8.99	−17.55, 0.46	12.80	9.18, 21.92	9.09	4.06, 12.35	0.94	0.84, 0.98
Pres(8s)avg2	Mean	28.85	31.74	29.85	30.15	Session 2-1	10.04	−4.51, 26.80	20.14	14.52, 33.95	10.08	3.88, 13.92	0.90	0.73, 0.96
	SD	11.53	11.55	8.95	10.74	Session 3-2	−1.08	−18.32, 19.80	28.12	20.10, 48.40	7.55	−1.97, 11.06	0.73	0.36, 0.90
						Session 3-2 ^#^	−10.01	−18.18, −1.02	12.31	8.84, 21.05	8.84	3.96, 12.01	0.95	0.84, 0.98
Pres(8s)avg3	Mean	29.40	31.41	30.58	30.46	Session 2-1	6.89	−4.03, −19.06	14.97	10.86, 24.88	10.47	4.64, 14.26	0.94	0.84, 0.98
	SD	11.64	11.18	9.49	10.81	Session 3-2	2.30	−15.27, 23.51	27.61	19.74, 47.46	7.62	−1.76, 11.12	0.74	0.38, 0.90
						Session 3-2 ^#^	−7.18	−13.84, 0.01	9.52	6.86- 16.14	9.00	4.24, 12.12	0.97	0.90, 0.99
*Force (N)*														
Force(8s)max	Mean	46.00	46.73	43.66	45.46	Session 2-1	5.62	−5.49, 18.03	15.46	11.21, 25.74	12.54	5.71, 17.06	0.96	0.88, 0.99
	SD	23.23	18.79	11.59	18.50	Session 3-2	0.02	−13.74, 15.97	21.10	15.20, 35.67	8.69	2.80, 12.14	0.86	0.63, 0.95
Force(8s)avg2	Mean	39.58	42.73	39.97	40.76	Session 2-1	10.87	0.00, 22.91	14.28	10.37, 23.69	13.86	6.47, 18.83	0.97	0.91, 0.99
	SD	18.89	17.65	13.12	16.74	Session 3-2	0.08	−15.80, 18.95	25.05	17.96, 42.78	10.25	3.29, 14.35	0.86	0.63, 0.95
Force(8s)avg3	Mean	40.52	43.15	39.83	41.17	Session 2-1	9.73	−0.15, 20.58	12.98	9.44, 21.47	13.28	6.25, 18.01	0.97	0.92, 0.99
	SD	20.20	17.97	11.66	17.00	Session 3-2	−0.65	−15.84, 17.28	23.95	17.19, 40.78	9.40	2.84, 13.19	0.85	0.61, 0.95

^#^ Outlier removed; * smallest effect from pure SD; CI: confidence interval: CV: coefficient of variation; SD: standard deviation.

**Table 6 sensors-23-03291-t006:** Measures of reliability in the less affected hand of participants with vision during complete grasp duration (8 s).

		Session 1	Session 2	Session 3	Mean 3 Sessions		Change in Mean (%)	90% CI for Change in Mean	CV (%)	90% CI for Change in CV	Smallest Effect (%) *	90% CI for Change in Smallest Effect	ICC	90% CI for Change in ICC
*Pressure (kPa)*														
Pres(8s)max	Mean	42.55	36.90	36.93	38.80	Session 2-1	−8.75	−27.21, 14.40	33.99	24.14, 59.37	6.84	−3.61, 10.57	0.61	0.16, 0.85
	SD	21.99	11.74	10.97	15.73	Session 3-2	1.07	−8.13, 11.19	13.15	9.56, 21.74	7.21	2.95, 9.86	0.91	0.76, 0.97
Pres(8s)avg2	Mean	40.30	35.57	34.98	36.95	Session 2-1	−7.96	−26.05, 14.55	32.73	23.27, 56.99	6.83	−3.40, 10.47	0.62	0.18, 0.86
	SD	20.03	11.62	10.72	14.73	Session 3-2	−0.57	−8.5, 8.11	11.43	8.33, 18.82	7.36	3.21, 10.01	0.93	0.81, 0.98
Pres(8s)avg3	Mean	40.22	34.82	34.86	36.63	Session 2-1	−9.00	−26.78, 13.08	32.47	23.10, 56.51	6.79	−3.37, 10.41	0.63	0.19, 0.86
	SD	20.53	10.86	10.69	14.76	Session 3-2	0.75	−7.53, 9.77	11.73	8.54, 19.33	7.19	3.09, 9.79	0.93	0.80, 0.98
*Force (N)*														
Force(8s)max	Mean	50.91	45.04	49.16	48.37	Session 2-1	−11.29	−25.01, 4.94	24.28	17.43, 41.38	7.49	−0.13, 10.76	0.78	0.45, 0.92
	SD	18.98	16.79	14.43	16.83	Session 3-2	11.52	−0.27, 24.70	15.55	11.27, 25.89	7.50	2.83, 10.34	0.89	0.70, 0.96
Force(8s)avg2	Mean	47.56	42.86	46.02	45.48	Session 2-1	−9.53	−23.91, 7.56	25.10	18.00, 42.87	7.14	−1.43, 10.37	0.75	0.40, 0.91
	SD	17.61	15.65	15.10	16.16	Session 3-2	8.27	−1.66, 19.21	13.25	9.64, 21.93	7.80	3.28, 10.64	0.92	0.78, 0.97
Force(8s)avg3	Mean	46.70	41.79	45.05	44.51	Session 2-1	−9.35	−23.71, 7.70	24.99	17.93, 42.68	7.22	−1.30, 10.46	0.75	0.41, 0.91
	SD	17.66	14.73	14.29	15.63	Session 3-2	8.81	−0.67, 19.20	12.52	9.11, 20.67	7.52	3.20, 10.25	0.92	0.79, 0.97

* Smallest effect from pure SD; CI: confidence interval: CV: coefficient of variation; SD: standard deviation.

**Table 7 sensors-23-03291-t007:** Summary of evaluation of measures of maximal tactile pressures/forces against reliability criteria in the affected hand of participants.

Sessions	Measures of Maximal Tactile Pressure/Force Over Complete Grasp Duration	Complete Grasp Duration (8 s) With Vision			Complete Grasp Duration (8 s) Without Vision			Measures of Maximal Tactile Pressure/Force Over Complete Grasp Duration	Plateau Phase (5 s) With Vision			Plateau Phase (5 s) Without Vision		
		Change in Mean (%)	CV (%)	ICC	Change in Mean (%)	CV (%)	ICC		Change in Mean (%)	CV (%)	ICC	Change in Mean (%)	CV (%)	ICC
Session 2-1	Pres(8s)max	good	x	very good	x	x	good	Pres(5s)max	very good	x	very good	x	x	good
Session 3-2		very good	x	x	good	x	good		very good	x	x	good	x	good
Session 3-2 ^#^		good	acceptable	very good					good	good	Very good			
Session 2-1	Pres(8s)avg2	good	x	very good	good	x	very good	Pres(8s)avg2	x	x	very good	good	x	very good
Session 3-2		very good	x	x	very good	x	good		very good	x	x	very good	x	good
Session 3-2 ^#^		good	acceptable	very good					good	good	Very good			
Session 2-1	Pres(8s)avg3	good	acceptable	very good	good	x	good	Pres(5s)avg3	good	x	very good	good	x	good
Session 3-2		very good	x	x	very good	acceptable	very good		very good	x	x	very good	acceptable	very good
Session 3-2 ^#^		good	acceptable	very good					good	good	Very good			
Session 2-1	Force(8s)max	good	x	very good	very good	x	very good	Force(5s)max	good	x	very good	very good	x	very good
Session 3-2		very good	x	good	good	acceptable	very good		very good	x	good	good	x	very good
Session 2-1	Force(8s)avg2	x	acceptable	very good	x	x	very good	Force(5s)avg2	x	acceptable	very good	x	x	very good
Session 3-2		very good	x	good	x	acceptable	very good		very good	x	good	good	acceptable	very good
Session 2-1	Force(8s)avg3	good	acceptable	very good	x	x	very good	Force(5s)avg3	good	acceptable	very good	x	x	very good
Session 3-2		very good	x	good	x	acceptable	very good		very good	very good	good	good	acceptable	very good

^#^ Outlier removed; x: unsatisfactory.

**Table 8 sensors-23-03291-t008:** Summary of evaluation of measures of maximal tactile pressures/forces against reliability criteria in the less affected hand of participants.

Sessions	Measures of Maximal Tactile Pressure/Force Over Complete Grasp Duration	Complete Grasp Duration (8 s) With Vision			Complete Grasp Duration (8 s) Without Vision			Measures of Maximal Tactile Pressure/Force Over Complete Grasp Duration	Plateau Phase (5 s) With Vision			Plateau Phase (5 s) Without Vision		
		Change in Mean (%)	CV (%)	ICC	Change in Mean (%)	CV (%)	ICC		Change in Mean (%)	CV (%)	ICC	Change in Mean (%)	CV (%)	ICC
Session 2-1	Pres(8s)max	good	x	x	good	good	very good	Pres(5s)max	good	x	x	good	good	very good
Session 3-2		very good	acceptable	good	very good	x	good		very good	x	good	very good	x	good
Session 2-1	Pres(8s)avg2	good	x	x	x	x	very good	Pres(5s)avg2	good	x	x	good	good	very good
Session 3-2		very good	acceptable	very good	good	good	very good		very good	x	good	very good	good	very good
Session 2-1	Pres(8s)avg3	good	x	x	good	x	very good	Pres(5s)avg3	x	x	x	good	good	very good
Session 3-2		very good	acceptable	very good	very good	good	very good		very good	x	good	very good	good	very good
Session 2-1	Force(8s)max	x	x	good	very good	x	very good	Force(5s)max	x	x	good	very good	good	very good
Session 3-2		x	x	good	very good	x	good		x	x	good	very good	good	good
Session 2-1	Force(8s)avg2	good	x	good	x	x	very good	Force(5s)avg2	x	x	good	good	good	very good
Session 3-2		good	acceptable	very good	good	x	good		x	x	very good	very good	good	good
Session 2-1	Force(8s)avg3	good	x	good	very good	x	very good	Force(5s)avg3	x	x	good	very good	good	very good
Session 3-2		good	acceptable	very good	very good	x	good		x	good	very good	very good	good	very good

x: Unsatisfactory.

**Table 9 sensors-23-03291-t009:** Average difference between consecutive sessions.

Group	Upper Limb		Average Difference between Consecutive Sessions		*p* Value
			Session 2-1	Session 3-2	
Stroke	Affected	Vision *	6.67	−7.45	0.07
		No vision	8.97	−3.73	0.19
	Less affected	Vision	−9.43	0.75	0.44
		No vision	7.10	−1.79	0.31

* Outlier removed.

**Table 10 sensors-23-03291-t010:** Concurrent validity of measures of maximal tactile pressures in stroke participants.

Upper Limb		Mean	SD	Mean Bias Raw	SD	Typical Error Raw	90% CI of Typical Raw Error	Typical Error (CV) %	90% CI of CV	Correlation (r)	90% CI of r ^a^	*p* Value
Affected	Maximal tactile pressures ^#^/kPa	33.70	8.28									
	Grip strength/kg	22.09	7.26	−2.20	5.91	8.18	5.96, 3.45	25.51	18.02, 45.32	0.37	−0.19, 0.75	0.002 *
Less affected	Maximal tactile pressures ^#^/kPa	40.22	20.53									
	Grip strength/kg	27.00	9.36	−2.51	8.89	17.97	13.10, 29.56	47.41	32.71, 89.35	0.64	0.17, 0.87	0.07

CI: confidence interval; CV: coefficient of variation; SD: standard deviation. ^#^ Based on Pres(8s)avg3; ^a^ based on log-transformed data. * Significant difference.

## Data Availability

The data presented in this study are available on request from the corresponding author. The data are not publicly available due to being used for additional analyses in other studies.

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
