# Peer review of "Measures of Maximal Tactile Pressures during a Sustained Grasp Task Using a TactArray Device Have Satisfactory Reliability and Concurrent Validity in People with Stroke"

_sensors, 2023, doi:10.3390/s23063291_

Round 1
Reviewer 1 Report
This study builds up on the systematic evaluation of the TactArray previously conducted in their laboratory and provides a preliminary investigation to determine reliability and limited external validity for measurement of maximal tactile pressures during a sustained grasp in people with stroke, which could provide a novel means of objective quantification of sustained grasp strength. The biggest problem was the small sample size, with only 11 participants, which makes the reliability of the measurements difficult to assess. Two suggestions are as follows:
1. As Jamar hand dynamometer is the most commonly used tool to assess grip strength. So give a comparison of the Grip strength between measured by TactArray device and by the Jamar hand dynamometer, so as to prove the accuracy of the measured result measured by TactArray device. If possible, include the pressure or force.
2. The serial number of the reference is repeated in the section of References.
Reviewer 2 Report
This work is entitled "Measures of maximal tactile pressures during a sustained grasp task using a TactArray device have satisfactory reliability and concurrent validity in people with stroke", which is to investigate the reliability and concurrent validity of measures of maximal tactile pressures and forces during a sustained grasp task using a TactArray device in people with stroke. However, I think that its sensors are not novel and significantly improved compared to previous reported materials and the writing of this manuscript seems like experiment report, it is not appropriate for Sensors. 1.The authors emphasized that the impact of stroke and vision on the maximum tactile measurement is the secondary goal of the article. However, it seems to have been emphasized from the earlier part of the article. 2.It is suggested that the author can select several similar works for comparison to highlight the advantages of this article. 3.Please provide your experimental data in detail, such as the difference between the grip strength data of stroke patients and normal people. It is better to present it in the form of figures. 4.The sensor and work content of this manuscript are not innovative. The authors used a large part of the paper to demonstrate the experimental process and analyze the experimental results. However, what applications this experimental result can bring to us are not emphasized, this referee would like to see a breakthrough in the application of the sensor.
Round 2
Reviewer 1 Report
The authors have explained my two concerns and made corresponding analysis and modification in the revision. but the biggest problem was the small sample size, which was raised last time, with only 11 participants, which makes the reliability of the measurement may be difficult to evaluate. so it is suggested to increase the sample size and measurement.
Reviewer 2 Report
I agree to accept this manuscript.
